# Erectile Dysfunction Severity: The Role of Glycometabolic Compensation and Antihyperglycemic Drugs

**DOI:** 10.3390/jcm11237214

**Published:** 2022-12-05

**Authors:** Virginia Zamponi, Giuseppe Defeudis, Francesca Federico, Antongiulio Faggiano, Rossella Mazzilli

**Affiliations:** 1Andrology Unit, Department of Clinical and Molecular Medicine, Sapienza University of Rome, 00189 Rome, Italy; 2Unit of Endocrinology and Diabetes, Department of Medicine, University Campus Bio-Medico di Roma, 00128 Rome, Italy

**Keywords:** erectile dysfunction, diabetes mellitus, GLP1, SGLT2, PDE5-i, antihyperglycemic drugs

## Abstract

Background: The aim of this study was to evaluate the prevalence of DM among patients with ED and the impact of glycometabolic compensation and antihyperglycemic treatment on ED severity. Methods: In total, 1332 patients with ED were enrolled. The diagnosis was performed through the International-Index-of-Erectile-Function questionnaire. ED severity was considered according to presence/absence of spontaneous erections, maintenance/achievement deficiency and response to PDE5-i. DM patients were clustered according to antihyperglycemic treatment: “metformin”/“insulin”/“old antihyperglycemic drugs”/“new antihyperglycemic drugs”. Results: The prevalence of DM patients was 15.8% (Group A, patients with ED and DM). Among these, the prevalence of spontaneous erections (21.0%) was lower than in the remaining patients (Group B, patients with ED without DM) (32.0%, *p* < 0.001). The prevalence of poor response to PDE5-i was lower in Group B (10.0%) than in Group A (35.0%, *p* < 0.001). Patients with good response to PDE5-i therapy showed lower HbA1c values than patients with poor/no response (6.6 ± 1.1% vs. 7.7 ± 1.9%, *p* = 0.02). The prevalence of absent response to PDE5-i was higher in patients treated with old antidiabetic drugs than in the population treated with new drugs (*p* = 0.03). Conclusion: The severity of ED and lower response to PDE5-i were higher in DM patients. A better glycometabolic profile, as well as new antihyperglycemic drugs, seem to have a positive effect on ED.

## 1. Introduction

Erectile dysfunction (ED) is defined as the recurrent inability to achieve or maintain an adequate erection and/or a noticeable decrease in erectile rigidity during partnered sexual activity. In order to meet the diagnostic criteria, these symptoms should persist for at least six months, occur on at least 75% of occasions and result in clinically significant personal distress. The disorder could be specified by severity and subtyped as either generalized or situational [1]. ED is characterized by its multidimensionality and consists of biological, intrapsychic and socio-relational components. The impairment of one of these factors can represent the primum movens, which can lead to the gradual disturbance of all the other components with possible negative effects on the quality of life (QoL) [2]. Several studies have explored the epidemiology of ED. A European multicenter study conducted on a population of men between 40 and 79 years old reported a prevalence of ED ranging from 6% to 64%, according to the different age subgroups and increasing with a prevalence of hypertension, heart disease and metabolic disease [3]. 

ED and Diabetes Mellitus (DM) share many risk factors, such as vasculopathy and neuropathy [4]. According to Corona et al., the incidence of ED in DM patients is 19.4% for the mild forms, 15.4% for the mild-moderate forms, 10.4% for the moderate ones and 21.6% for the severe ones [5]. However, the epidemiological data on the association of ED with DM vary in different studies according to age, duration of DM and the diagnostic criteria and methodology. A meta-analysis showed a prevalence of ED in patients with DM of 52.5%, and patients with DM have a 3.5 times greater risk of developing ED than patients without DM. The same meta-analysis highlighted that the age-related risk of ED is doubled in DM compared to the healthy male population [6]. ED should also be considered an early marker of atherosclerosis, playing a key role in predicting the onset of cardiovascular events (CVOTs) [7]. In fact, although ED is a multifactorial condition, in patients with DM, the risk of a prevalent organic etiology is higher and should be carefully taken into account [4]. For this reason, is mandatory for clinicians to investigate the occurrence of ED during the DM complication screening. Along with the old antihyperglycemic drugs (insulin, metformin, sulfonylureas, glinides and thiazolidinediones), in recent decades the following new antihyperglycemic drugs have been introduced for the treatment of DM: Dipeptidyl-peptidase-4 inhibitors (DPP4-i), glucagon-like peptide- 1 receptor agonists (GLP1-Ras) and sodium-glucose cotransporter-2 inhibitors (SGLT2-i). These medications have been shown to have a cardio- and vaso-protective effect [8]. To date, few studies have evaluated the impact of new antihyperglycemic drugs on ED. Although the data are promising, there is a lack of human and large case studies [9]. Considering ED treatment, Phosphodiesterase 5 inhibitors (PDE5-i) are the first-line treatment; however, in patients with uncontrolled DM and/or complicated DM, they could have poor or no response on erectile function [10].

The aim of this study was to evaluate the prevalence of DM in a large population of patients with ED and the impact of glycometabolic compensation, as well as the type of antihyperglycemic treatment on the severity of ED.

## 2. Materials and Methods

### 2.1. Participants

In the present monocentric retrospective study, all patients affected by ED and referred to the Andrology Unit of the Sant’Andrea University Hospital in Rome from 2013 to 2020 were enrolled. The experimental group was composed of 210 subjects affected by ED and DM (Group A), while the remaining patients with ED and without DM represented the Group B. 

All participants met the following inclusion criteria: age 18–75 years; presence of DM type 2 (T2DM) for the Group A. 

For all participants, the exclusion criteria were the use of antiandrogens, narcotics and stimulants; the presence of uncontrolled acute disease: psychiatric diseases with recent consumption of psychiatric drugs; uncontrolled endocrine diseases; individuals undergoing prostate surgery; and active or previous oncological diseases excluding basal cell carcinoma. 

### 2.2. Procedures

All patients underwent a detailed medical and sexological history collection and a physical examination. The diagnosis of ED was performed through the administration of Simplified International Index of Erectile Function (IIEF-5) questionnaire. 

In patients affected by DM, the following aspects were considered: (a) time of onset and duration of DM; (b) glycosylated hemoglobin values (HbA1c); (c) comorbidities: hypertension (yes/not) and dyslipidemia (yes/not); (d) treatment of DM, divided into the following categories: “metformin”, “insulin”, “old antidiabetic drugs” (including sulfonylureas, glitazones and 5alpha glucosidase inhibitors) and “new antidiabetic drugs” (including SGLT2-i, GLP1-Ras, DPP4-i).

For the evaluation of severity of ED, according to a previous study [11], the following parameters were considered: (a) achievement or maintenance of erections (yes/no; this parameter is also evaluated by questions n.2 and n.3 of the IIEF-5 [12]); (b) presence or absence of spontaneous erections (that means an increase in blood flow in the corpora cavernosa and a contraction of the ischiocavernosus and bulbospongiosus muscles [13]); (c) response to PDE5-i drugs (good, poor or absent).

All patients underwent blood sampling at 8:00 a.m. to measure: luteinizing hormone (LH), follicle stimulating hormone (FSH), thyroid stimulating hormone (TSH), total testosterone (TT), estradiol, prolactin, Sex Hormone Binding Globulin (SHBG), HbA1c and lipid profile (total cholesterol, HDL, LDL, triglycerides). Chemiluminescence microparticle immunoassay (CMIA) and immunoassay (CLIA) were used.

### 2.3. Statistical Analysis

Mean ± standard deviation was calculated for all measured variables. 

Continuous variables were described as absolute values, mean ± standard deviation (SD). Categorical variables were described as absolute numbers and percentage (%). The Fischer test for the analysis of categorical variables and the T-Test for continuous variables were used. 

A *p* value < 0.05 was considered statistically significant. 

Statistical analyses were carried out on the software R version 2.14.2 (Free software Foundation Inc., Boston, MA, USA).

### 2.4. Ethics

The study adhered to the Hospital’s Ethics Committee guidelines and to the Ethical Principles for Medical Research Involving Human Subjects as adopted at the 18th WMA General Assembly, Helsinki, Finland, June 1964; amended by the 55th WMA General Assembly, Tokyo, Japan, October 2004; and subsequent modifications when enforced (last, Fortaleza, Brazil, October 2013).

The study was approved by the Ethics Committee of the Hospital (Protocol n. RIF. CE 6559_2021)

## 3. Results

### 3.1. Comparison between Patients with ED with and without DM

A total of 1782 patients with ED were evaluated. Among them, 450 patients did not meet the inclusion criteria. Thus, the final sample size included 1332 patients. Of these, 210 (15.8%) were affected by DM (Group A), while the remaining number of patients with ED was 1122 (84.2%) (Group B). Table 1 shows the basal characteristics of the two groups.

### 3.2. Age and Duration of DM

The mean age ± SD of Group A was higher than Group B (61.0 ± 10.0 years vs. 53.0 ± 13.9 years; *p* < 0.001). The duration of DM was 18.0 ± 10.0 years.

HbA1c values were higher in Group A than in Group B (7.3 ± 1.5% vs. 5.5 ± 0.43%; *p* < 0.001) (Table 1).

### 3.3. Lipid and Hormonal Profile

Statistically significant differences were found between two groups for the triglycerides value, which was higher in Group A than Group B (238.0 ± 128.0 vs. 150.0 ± 75.0, *p* = 0.02), and for HDL, which was lower in Group A compared to group B (31.0 ± 9.2 vs. 50.0 ± 11.0, *p* = 0.02). There were no statistically significant differences in the mean ± SD values of total and LDL cholesterol.

The mean ± SD values of LH were higher in Group A compared to Group B (7.3 ± 5.0 vs. 5.0 ± 4.2; *p* = 0.01), as well as the mean ± SD value of TSH (3.2 ± 2.8 vs. 1.7 ± 1.4; *p* = 0.049). On the contrary, the mean ± SD value of estradiol was significantly lower in Group A than in Group B (14.3 ± 3.5 vs. 32.5 ± 13.6; *p* = 0.04). No statistically significant difference between the two groups were found in total testosterone, FSH, prolactin and SHBG values (Table 1).

### 3.4. Comorbidity

The prevalence of dyslipidemia and hypertension was higher in the Group A (40.0% and 44.0%, respectively) compared to Group B (13.0% and 22.0%) (*p* < 0.001) (Table 1). 

### 3.5. Severity of ED

Among patients of Group A, the prevalence of spontaneous erections (21.0%) was lower than Group B (32.0%, *p* < 0.001); on the other hand, the prevalence of sporadic or absent spontaneous erections was higher in Group A (79.0%) than in Group B (68.0%) (*p* < 0.001).

No significant differences were observed regarding the prevalence of difficulty in the achievement or maintenance the erection between Group A (48.0% and 52.0%, respectively) and Group B (38.0% and 62.0%) (*p* = 0.09).

Among the patients of Group A, 105 received PDE5-i. Of them, 22.0% reported a good response, 35.0% a poor response and 30.0% no response. In Group B, 456 were treated with PDE5-i. Of them, 45.0% reported a good response, 10.0% a poor response and 24.0% no response. The prevalence of good response to PDE5-i was statistically higher in Group B (45.0%) than in Group A (22.0%, *p* < 0.001); conversely, the prevalence of poor response was significantly lower in Group B (10.0%) than in Group A (35.0%, *p* < 0.001).

### 3.6. Impact of Glucose Control and Antidiabetic Therapy on the Severity of Erectile Dysfunction in Diabetic Patients

#### 3.6.1. Glucose Control and Severity of ED

Among group A, patients with good response to PDE5-i therapy showed HbA1c value lower than patients with poor and no response to PDE5-i (6.6 ± 1.1% vs. 7.7 ± 1.9%, *p* = 0.02) (Table 2).

Furthermore, the mean ± DS of HbA1c values were higher in patients with sporadic or absent spontaneous erections than in the group of patients with present and frequent spontaneous erections (7.4 ± 1.6% vs. 7.0 ± 1.3%). However, significant differences were found between the two groups (*p* = 0.55) (Table 2). Then, although the mean ± SD values of HbA1c were higher in patients with difficult in the achievement of erection than in the group of patients with difficult in the maintenance (7.5 ± 1.5% vs. 6.8 ± 1.1%), no statistically significant difference (*p* = 0.06) between two groups were found (Table 2).

Patients were also divided according to DM compensation, namely “adequate glycemic control” (i.e., HbA1c < 7%) and “inadequate glycemic control” (i.e., HbA1c > 7%). The prevalence of patients with difficult in the achievement of an erection was higher in the group with “inadequate glycemic control” (68.2%) than “adequate glycemic control” (37.0%, *p* = 0.045).

No significant differences were observed for the other severity parameters.

#### 3.6.2. Antihyperglycemic Treatment and ED Severity

No statistically significant differences were observed between the classes of antihyperglycemic treatment and presence/absence of spontaneous erections, as well as difficulty in their achievement or maintenance.

Conversely, the prevalence of absence of response to PDE5-i was higher in patients treated with older generation of antihyperglycemic drugs than in the population treated with new drugs (*p* = 0.03).

### 3.7. Impact of Comorbidities on the Severity of Erectile Dysfunction in Diabetic Patients

Considering Group A, no significant differences were observed regarding the prevalence of spontaneous erections between patients with or without hypertension (*p* = 0.1171). The same results were observed regarding the prevalence of difficulty in the achievement or maintenance the erection (*p* = 0.5587) and for the prevalence of good or absent responses to PDE5-i (*p* = 8035). 

Similarly, no significant differences were observed in Group A regarding the prevalence of spontaneous erections between the patients with or without dyslipidemia (*p* = 0.5646). The same results were obtained regarding the prevalence of difficulty in the achievement or maintenance the erection (*p* = 0.4498) and for the prevalence of good or absent responses to PDE5-i (*p* = 6029). 

## 4. Discussion

In the present retrospective study, 1332 patients with ED referred to the Andrology Unit of Sant’Andrea University Hospital in Rome from 2013 to 2020 were evaluated. Of them, 210 were affected by DM (Group A), while the Group B was composed by 1122 patients with ED but without DM. Overall, the prevalence of DM in ED population was 15.8%. This result is in accordance with the data from literature [14,15]; specifically, the prevalence of ED in a diabetic population was 19.5%; the prevalence also increased with the age of the patients [14]. 

As expected, in the present study, from the comparison of the basic characteristics of the two populations, glycometabolic compensation was worse in Group A than in Group B, with higher values of HbA1c and triglycerides and significantly lower HDL levels. Several studies highlighted the atherogenic role of triglycerides and its implication in the pathogenesis of ED [16,17]. Dyslipidemia is associated with increased lipid peroxidation, which predisposes one to the development of atherosclerosis [18]. Furthermore, superoxide radicals could impair the relaxation of the cavernous smooth muscle, resulting in ED [18]. 

Although no statistically significant differences were observed in total testosterone values between the two groups, the LH value was significantly higher in the group of patients with DM. These results differ from most data in the literature. In fact, several cross-sectional studies demonstrated that up to 40% of men with T2DM exhibit a hypogonadism condition associated with inappropriately low gonadotropin levels [4,19]. However, the functional hypogonadism could be due to the negative feedback on the hypothalamus–pituitary–gonads axis acting by higher estradiol levels for increased activity of aromatase enzyme on visceral fat [20]. Furthermore, the reduced SHBG levels is frequently observed in patients with DM. Low SHBG levels are inversely correlated with HbA1c values and with the development of DM [21]. In the present study, higher LH levels could suggest a subclinical primary hypogonadism due to initial testis impairment. Indeed, several studies demonstrated that both hyperglycemia and hyperinsulinemia could damage the blood–testis barrier, which can lead to Leydig cell apoptosis and decrease spermatogenesis [22,23]. 

Considering comorbidities, a higher prevalence of hypertension and dyslipidemia was found in Group A than in Group B. These data agree with the literature: ED and DM share many risk factors and are associated with pathologies that may underline endothelial dysfunction [4]. High blood pressure values could have a negative impact on the physiology of erections due to possible damage to the penile vessels [24]. These conditions contribute to the reduction in blood flow and endothelial impairment with consequent reduction in NO production [25]. Moreover, patients with severe penile vessels atherosclerosis may experience reduced or absent response to PDE5-i [26]. Dyslipidemia also contributes to endothelial damage through increased atherogenic risk [27]. For this reason, dyslipidemia is considered an independent risk factor for the development of ED in patients with DM [28]. However, although Group A showed significantly higher prevalence of hypertension and dyslipidemia, we found no significant differences of ED severity when clustering patients with DM according to the presence/absence of these comorbidities. These findings suggest that, in this setting of patients, the glycemic control worsened the sexual function more than hypertension and dyslipidemia.

For the evaluation of the impact of DM on the severity of ED, the following parameters were considered: presence or absence of spontaneous erections, difficulties in achieving or maintaining the erection and the response to PDE5-i (good, poor or absent). We found a prevalence of good response to PDE5-i that was statistically significantly higher in Group B than in Group A. These data suggest that patients with DM could have more severe ED than subjects without DM. These results are comparable to what is reported in the literature. Several authors showed that men with DM have a higher risk of developing ED than healthy subjects and that glucose compensation, DM duration and age have a significant impact on ED severity [4]. To assess the impact of metabolic compensation on male sexual function in patients with DM, we compared the three severity parameters of the ED with the HbA1c values, as well as the adequate/inadequate glycemic control. In the present study, a higher mean value of HbA1c in patients with poor and absent response to PDE5-i therapy was found, and the difficulty in the achievement of an erection was higher in patients with inadequate glycemic control.

Furthermore, the trend of results also showed higher mean values of HbA1c in patients with absent spontaneous erections than in the group of patients with the presence of spontaneous erections. The same trend was found in patients with ED in the achievement of erections, whom had higher mean values of HbA1c than the patients with ED in maintenance. These data suggest a key role of glycometabolic compensation on ED severity. 

Several studies demonstrated that hyperglycemia and insulin resistance could lead to a reduction in NO synthase activity, resulting in penile vasodilation impairment [29]. These data are also confirmed in a study conducted on mice with induced DM, which showed that insulin resistance and hyperglycemia reduce the expression of both endothelial and neuronal NO synthase (eNOS, nNOS) [30]. Finally, hyperglycemia induces the production of advanced glycation end products (AGEs) and reactive oxygen species (ROS) that play a key role on peripheral neuropathy and endothelial damage [31,32].

Previous studies highlighted that the glycometabolic compensation could also affect the efficacy of PDE5-i; despite PDE5-i being considered the first-line therapy for ED, their lower efficacy in patients affected by DM was demonstrated [33,34]. The reduced effectiveness of PDE5-i in subjects with DM is due to the general endothelium dysfunction, as well as the onset of hypogonadism and concomitant antidiabetic medication [35]. All these factors could impair the molecular pathway that led to NO and GMPc reductions [36]. Our results confirmed the data present in literature.

Finally, the possible impacts of old and new antihyperglycemic drugs on the severity of ED were investigated. No statistically significant differences were found from the comparison between old and new antihyperglycemic drugs and the presence or absence of spontaneous erections. The same results were obtained from the comparison between old and new antihyperglycemic drugs and difficulties in the achievement or maintenance of erections. Otherwise, a statistically significant difference in the comparison between old and new antidiabetic drugs and the response to PDE5-i was found, showing a higher prevalence of no response to PDE5-i therapy in patients treated with old antidiabetic therapies compared to patients treated with new antidiabetic therapies. 

This result was in accordance with other reports, which highlighted that the IIEF-5 score is higher in patients on therapy with new antidiabetic drugs, in particular with GLP1-Ras, compared to the scores obtained by patients treated with insulin or old antidiabetic drugs [9,37]. So far, the studies conducted on the impact of GLP1 and SGLT2i on male sexual function have mostly been conducted on mouse models, but the results seem to be encouraging [38,39]. Giagulli et al., in a recent case–control study, demonstrated that the addition of Liraglutide to Metformin in diabetic men improved sexual function compared to the group treated with Metformin alone [40]. These studies suggested that the new antihyperglycemic drugs could improve erectile function through their cardioprotective action [41]. 

The retrospective design of the present study represents a major limitation, not allowing us to evaluate the clinical response to pharmacological treatments in patients with hypertension and dyslipidemia. Moreover, the small number of patients treated with new antihyperglycemic drugs does not allow for a statistically relevant comparison between traditional and more recent antidiabetic treatments.

## 5. Conclusions

This study confirms the high prevalence of DM in the andrological population with ED. Patients with DM seem to have a greater severity of ED and minor response to PDE5-i than patients without DM. The glycometabolic compensation of DM appears to have a significant impact on ED severity, in particular on response to PDE5-i therapy, as well as the new antihyperglycemic drugs, which would seem to have a protective role on male sexual function. This aspect should be pointed out in the clinical setting, since a targeted counselling on the relation between DM and ED may motivate patients to reacha good glycemic control.

Further prospective and larger studies are needed to investigate the impact of DM on ED severity and the role of new antihyperglycemic drugs on male sexual function.

## Figures and Tables

**Table 1 jcm-11-07214-t001:** Comparison of basal features between patients with DE and DM (Group A) and patients with DE (Group B).

Basal Features	Total Group	Group A	Group B	*p*-Value
Age (mean ± SD, years)	54.0 ± 19.1	61.0 ± 10.0	53.0 ± 13.9	<0.001
Duration of DM (mean ± SD, years)	/	18.0 ± 10.0	/	/
HbA1c (mean ± SD, %)	5.6% ± 0.5	7.3% ± 1.5	5.5% ± 0.43	<0.001
Testosterone (mean ± SD, ng/mL)	4.4 ±1.6	4.7 ± 1.4	4.5 ± 1.6	0.06
FSH (mean ± SD, mIU/mL)	8.9 ± 42.9	9.6 ± 8.4	7.1 ± 7.8	0.09
LH (mean ± SD, mIU/mL)	5.2 ± 4.3	7.3 ± 5.0	5.0 ± 4.2	0.01
SHBG (mean ± SD, nmol/L)	13.0 ± 29.8	16.6 ± 16.7	30.0 ± 20.7	0.56
PRL (mean ± SD, ng/mL)	9.9 ± 6.0	10.8 ± 5.7	10.0 ± 6.1	0.84
Estradiol (mean ± SD, ng/mL)	32.0 ± 13.5	14.3 ± 3.5	32.5 ± 13.6	0.04
TSH (mean ± SD, mU/L)	1.7 ± 1.1	3.2 ± 2.8	1.7 ± 1.4	0.049
Total cholesterol (mean ± SD, mg/dL)	211.0 ± 41	207.0 ± 31.1	211.0 ± 42.2	0.73
LDL (mean ± SD, mg/dL)	142.0 ± 38.2	192.0 ± 93.3	138.0 ± 31.2	0.56
HDL (mean ± SD, mg/dL)	48.0 ± 12	31.0 ± 9.2	50.0 ± 11.0	0.02
Triglycerides (mean ± SD, mg/dL)	186.0 ± 141	238.0 ± 127.6	150.0 ± 75.4	0.02
Dyslipidemia (n, %)	224, 17.0%	83, 40.0%	141, 13.0%	<0.001
Hypertension (n, %)	341, 26.0%	92, 44.0%	247, 22.0%	<0.001

SD = Standard Deviation; DM = Diabetes Mellitus; HbA1c = glycated hemoglobin; FSH = Follicle Stimulating Hormones; LH = Luteinizing Hormone; SHBG = Sex Hormone Binding Globulin; PRL = Prolactin; TSH = Thyroid Stimulating Hormone; LDL = Low Density Lipoprotein; HDL = High Density Lipoprotein.

**Table 2 jcm-11-07214-t002:** Comparison of Hba1c and ED severity in Group A.

ED Severity		Hba1c (Mean ± DS, %)	*p*-Value
Spontaneous erections	Present and frequent	7.0 ± 1.3	0.55
	Sporadic or absent	7.4 ± 1.6	
Erectile dysfunction	Achievement	7.5 ± 1.5	0.06
	Maintenance	6.8 ± 1.1	
PDE5-i response	Good	6.6 ± 1.1	0.02
	Poor/Absent	7.7 ± 1.9	

SD = Standard Deviation; ED = Erectile Dysfunction; HbA1c = glycated hemoglobin; PDE5-i = Phosphodiesterase-5 inhibitors.

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
