# Peer review of "Erectile Dysfunction Severity: The Role of Glycometabolic Compensation and Antihyperglycemic Drugs"

_jcm, 2022, doi:10.3390/jcm11237214_

Round 1

Reviewer 1 Report

The topic of this manuscript is very relevent to clinical practice. DE is often overlooked as a complication of Diabetes and Diabetes in not always considered in visits for DE.

One of the major strengths of this study is its sample size. However, my major remark regards statistical analysis. Retrospective data can have many limitations which can be overcome by statistical methods. For instance, the Diabetes population you considered is older and has a very high LDL, which might play an important role in DE, in addition to diabetes. I suggest you account for these factors, as well as statin use among participants in both groups. When comparing ED severity among diabetic patients, I suggest you divide the group into "adeguate glycemic control", i.e. HbA1c <7%, and "inadeguate glycemic control", i.e. HbA1c > 7%. This division will probably give more value to your results.

I also have some minor suggestions that might help improve the manuscript:

1. "Figures, tables and schemes" is labeled as point 3.2 but it comes after paragraph 3.6.2. What does 3.2 refer to?

2. Table 1: group A label has an extra 210

3. Table 1 and Table 2: insert a legend where you explain all acronyms present in the tables.

4. Reference 8: change "Association American Diabetes" into "American Diabetes Association"

Author Response

- One of the major strengths of this study is its sample size. However, my major remark regards statistical analysis. Retrospective data can have many limitations which can be overcome by statistical methods. For instance, the Diabetes population you considered is older and has a very high LDL, which might play an important role in DE, in addition to diabetes. I suggest you account for these factors, as well as statin use among participants in both groups. When comparing ED severity among diabetic patients, I suggest you divide the group into "adeguate glycemic control", i.e. HbA1c <7%, and "inadeguate glycemic control", i.e. HbA1c > 7%. This division will probably give more value to your results.

R: Thank you for your suggestion.

Following your indications, we performed the following evaluations:

Results:

“Patients were also divided according to DM compensation, namely "adequate glycemic control" (i.e. HbA1c <7%) and "inadequate glycemic control”( i.e. HbA1c > 7%). The prevalence of patients with difficult in the achievement of erection was higher in the group with "inadequate glycemic control” (68.2) than "adequate glycemic control" (37%). No significant differences were observed for the other severity parameters.”

Discussion:

“To assess the impact of metabolic compensation on male sexual function in patients with DM, we compared the three severity parameters of the ED with the HbA1c values as well as the adequate/inadequate glycemic control. In the present study a higher mean value of HbA1c in patients with poor and absent response to PDE5-i therapy was found, and the difficulty in the achievement of erection was higher in patients with inadequate glycemic control.”

We added a dedicated section “Severity of DE and comorbidity” in the results section:

“3.7 Impact of comorbidities on the severity of erectile dysfunction in diabetic patients

Considering Group A, no significant differences were observed regarding the prevalence of spontaneous erections between patients with or without hypertension (p=0.1171). The same results were observed regarding the prevalence of difficulty in the achievement or maintenance the erection (p=0.5587) and for the prevalence of good or absent response to PDE5-i (p=8035).

Similarly, no significant differences were observed in Group A regarding the prevalence of spontaneous erections between the patients with or without dyslipiemia (p=0.5646). The same results were obtained regarding the prevalence of difficulty in the achievement or maintenance the erection (p=0.4498) and for the prevalence of good or absent response to PDE5-I (p=6029). “

We discussed these results as follows:

“However, although Group A showed significant higher prevalence of hypertension and dyslipidemia, we found no significant differences of ED severity clustering patients with DM according to presence/absence of these comorbidities. These findings suggest that, in this setting of parients, the glycemic control worsened the sexual function more than hypertension and dyslipidemia.”

Finally, a sentence was also added in the ”limitations” section.  

I also have some minor suggestions that might help improve the manuscript:

1. "Figures, tables and schemes" is labeled as point 3.2 but it comes after paragraph 3.6.2. What does 3.2 refer to?

R: thank you. We corrected the mistake.

2. Table 1: group A label has an extra 210

R: thank you. We deleted it.

3. Table 1 and Table 2: insert a legend where you explain all acronyms present in the tables.

R: thank you. We added them.

4. Reference 8: change "Association American Diabetes" into "American Diabetes Association"

R: thank you. We corrected it.

Reviewer 2 Report

Interesting article

- in the introduction reiterated ED mechanism is multifactorial

- there may be atherosclerotic lesions that make drug therapies refractory. In these cases there are dedicated endovascular treatments (PMCID: PMC8074129)

- Methodologically simple but solid article

- Group A and Group B merge. Reiterate that group A is that of diabetic patients. Also in the table.

- You can not make a correlation between glycated hemoglobin and IIEF-5

- In terms of "behavioral science", the patient could be incentivized to improve his glycemic profile to improve his sexual performance

- well written, perhaps a little more synthesis is needed. And a more captivating style/writing

Author Response

- in the introduction reiterated ED mechanism is multifactorial.

R: Thank you for the observation. The concept of the ED as a multifactorial condition has been remarked in the introduction section:

“In fact, although ED is a multifactorial condition, in patients with DM the risk for a prevalent organic etiology is higher and should be carefully taken into account”

- there may be atherosclerotic lesions that make drug therapies refractory. In these cases there are dedicated endovascular treatments (PMCID: PMC8074129)

R: thank you for your suggestion.

We discussed this aspect as follows: “Moreover, patients with severe penile vessels atherosclerosis may experience reduced or absent response to PDE5-i [ Sangiorgi G, et al].” (see discussion section)

- Methodologically simple but solid article

R: thank you.

- Group A and Group B merge. Reiterate that group A is that of diabetic patients. Also in the table.

Thank you for the observation. We reiterated this aspect in table1, in the abstract, and in “participant” section, so that is easier to understand for the reader.

- You can not make a correlation between glycated hemoglobin and IIEF-5

R: You are right. We evaluate the severity of ED with presence/absence of spontaneous erections, maintenance/achievement deficiency and response to PDE5-i. We better specify the sentence “a) achievement or maintenance of erections (yes/not; this parameter is also evaluated by the questions n.2 and n.3 of the IIEF-5)”

- In terms of "behavioral science", the patient could be incentivized to improve his glycemic profile to improve his sexual performance

R: Thank you for tour suggestion. We highlighted this aspect in the conclusions section, as follows “This study confirms the high prevalence of DM in the andrological population with ED. Patients with DM seem to have a greater severity of ED and minor response to PDE5-i than patients without DM. The glycometabolic compensation of DM appears to have a significant impact on ED severity, in particular on response to PDE5-i therapy, as well as the new antihyperglycemic drugs, which would seem to have a protective role on male sexual function. This aspect should be pointed out in the clinical setting, since a targeted counselling on the relation between DM and ED may motivate patients reaching a good glycemic control. Further prospective and larger studies are needed to investigate the effect of the impact of DM on ED severity and the role of new antihyperglycemic drugs on male sexual function.”

- well written, perhaps a little more synthesis is needed. And a more captivating style/writing

R: thanks for this suggestion. We revised the style/writing.